# Selective Laser Melting Additive Manufactured Tantalum: Effect of Microstructure and Impurities on the Strengthening-Toughing Mechanism

**DOI:** 10.3390/ma16083161

**Published:** 2023-04-17

**Authors:** Fengjun Lian, Longqing Chen, Changgui Wu, Zhuang Zhao, Jingang Tang, Jun Zhu

**Affiliations:** 1College of Physics, Sichuan University, Chengdu 610065, China; lian_fengjun@163.com (F.L.); wuchanggui1314@163.com (C.W.); 2Key Laboratory of Radiation Physics and Technology of Ministry of Education, Institute of Nuclear Science and Technology, Sichuan University, Chengdu 610064, China; chenlongqing@scu.edu.cn; 3Institute of Machinery Manufacturing Technology, China Academy of Engineering Physics, Mianyang 621900, China; zhuangphd2018@163.com

**Keywords:** tantalum, selective laser melting, impurity, strength, toughness

## Abstract

The balance between the strength and the toughness of pure tantalum (Ta) fabricated with selective laser melting (SLM) additive manufacturing is a major challenge due to the defect generation and affinity for oxygen and nitrogen. This study investigated the effects of energy density and post-vacuum annealing on the relative density and microstructure of SLMed tantalum. The influences of microstructure and impurities on strength and toughness were mainly analyzed. The results indicated that the toughness of SLMed tantalum significantly increased due to a reduction in pore defects and oxygen-nitrogen impurities, with energy density decreasing from 342 J/mm^3^ to 190 J/mm^3^. The oxygen impurities mainly stemmed from the gas inclusions of tantalum powders, while nitrogen impurities were mainly from the chemical reaction between the molten liquid tantalum and nitrogen in the atmosphere. The proportion of <110> texture decreased after vacuum-annealing at 1200 °C, while that of the <100> texture increased. Concurrently, the density of dislocations and small-angle grain boundaries significantly decreased while the resistance of the deformation dislocation slip was significantly reduced, enhancing the fractured elongation up to 28% at the expense of 14% tensile strength.

## 1. Introduction

Due to the exceptional properties of a high melting point and outstanding corrosion resistance [1], tantalum (Ta) and its alloys were extensively utilized in industries such as aerospace, nuclear industry, electronics, chemistry, etc. In addition, the high biocompatibility of tantalum provides high potential, especially for application in bio-implants and medical devices [2]. Conventionally, tantalum components are generally fabricated by powder metallurgy, powder injection molding, or electron beam melting techniques. However, as required structures become more and more complex, traditional reductions in material processing technologies have a dilemma in manufacturing an ideal tantalum part. In contrast, additive manufacturing technology presents “near net shape forming” through layer-by-layer accumulation [3,4,5]. Selective laser melting (SLM) is an additive manufacturing technology using laser as energy and powders as raw material [6,7]. Complex structure parts can be prepared by SLM. In addition, the forming accuracy is higher, and the microstructure is finer than that of other additive manufacturing technologies such as wire + arc additive manufacturing (WAAM), laser melting deposition (LMD), etc. [8]. Therefore, the SLM technique has great potential in manufacturing tantalum components with excellent surface quality, high accuracy, and complex structures.

However, SLMed tantalum presents high preferential texture and impurities attributed to its high-temperature gradient and oxygen-nitrogen affinity, which, in turn, could hinder its engineering application. Livescu Veronica et al. fabricated high-density pure tantalum with a columnar crystal and texture orientation, mainly being <111> and <100> [9]. It has been proved that the microstructure texture orientation is related to mechanical properties closely. In addition, interstitial impurities exhibit a great solid solution-strengthening effect for BCC tantalum, which affects strength and toughness significantly. According to the above facts, the balance between the strength and toughness of pure Ta prepared by SLM is mainly concerned with the optimization of pre-process parameters, such as laser power, scanning speed, preheating temperature, etc. A relative density of 96.9% was achieved, with the energy density being 241 J/mm^3^ using SLM by Zhou L B. In addition, it was found that the strength of SLMed Ta was higher than that of Ta fabricated with powder metallurgy [10]. However, as shown in Table 1, SLMed pure Ta had high strength but poor toughness compared to traditional manufacturing methods. Therefore, a balance between strength and toughness was required.

Microstructure and texture orientation especially effectively affect the mechanical properties of SLMed pure Ta [11]. Song Changhui et al. found that excessive energy density resulted in pore defects, while a too low energy density caused unfused holes and cracks, which could reduce the tensile mechanical performance of SLMed Ta [12]. Lore Thijs et al. investigated the effect of scanning strategies on texture orientation and mechanical properties. It has been shown that that most of the grains are oriented to <111> along the manufacturing direction [13]. Moreover, the mechanical properties of SLMed Ta are closely related with interstitial impurities.

Interstitial impurities from the forming chamber and metal powders had a significant effect on the solid solution strengthening of SLMed pure Ta due to the interaction between the dislocation and lattice distortion caused by impurities. Marinelli G found that the greater the oxygen content, the greater the hardness and yield strength of the arc additive manufactured (WAAM) Ta. However, the high oxygen content could result in micropores, which lead to elongation reduction [14]. Le G M fabricated pure Ta using a laser melting deposition (LMD) technology with coarse and fine spherical powders. The oxygen and nitrogen content of pure tantalum prepared with coarse spherical tantalum powders increased by 123 ppm and 60 ppm, which was far lower than that with fine spherical tantalum powders and resulted in relatively low hardness and high plasticity [15]. However, further research is still needed to understand the effect of oxygen and nitrogen impurities on the mechanical performance of SLMed Ta.

The microstructures and mechanical properties of SLMed metal and alloys could be influenced by post-heat treatments. Lucia-Antoneta Chicos et al. first investigated the microstructure, microhardness, and weight of SLMed Ti-6Al-4V heat treated by concentrated solar energy in the horizontal and vertical solar furnace. The α′ microstructure was transformed into a mixture of α and β below a heat temperature of 995 °C, while the original α′ microstructure was transformed into a lamellar α + β one with the formation of α colonies. The microhardness values of the SLMed Ti-6Al-4V after heat treatment had a higher dispersion than those untreated in both the cross-section and the horizontal section [16]. In addition, they also pronounced that the decomposition of α′ martensite after heat treatment in the solar furnace was influenced by temperatures and was different between the top and middle areas [17]. Alaei Moeen et al. found that dislocation density, microhardness, and yield strength decreased, but plasticity increased with the temperature being elevated from 1100 °C to 1400 °C for 4 h [18]. Zhou Libo et al. studied a three-stage heat treatment for SLMed Ta, with 1800 °C for 2 h, 1600 °C for 2 h, and 1400 °C for 2 h subsequently, which resulted in a transformation from columnar crystals to equiaxed crystals, including a sub-grain dislocation microstructure and tensile elongation that increased from 2% to 12% [19]. Therefore, the toughness of SLMed Ta could be further enhanced with post-heat treatments. Compared with powder metallurgy, significant differences exist in the heat treatment parameters of pure tantalum prepared by SLM, which mainly depends on the different microstructures and melting processes.

In this study, pure Ta samples were manufactured while the process parameters were optimized. The effect of energy density and post vacuum annealing on the relative densities and microstructure of SLM tantalum was investigated. The effect of the microstructure and impurities on the mechanism of strength and toughness was mainly analyzed.

**Table 1 materials-16-03161-t001:** Tensile properties of pure Ta fabricated by different processing techniques.

Condition	Ultimate Tensile Strength (MPa)	Elongation (%)
Powder metallurgy [10]	310	30
Hot isostatic pressing [20]	415–747	27–43
Electron beam melted [10]	205	40
Cold-rolled [14]	265	62
Annealed [10]	200–390	20–50
Wrought [1]	178	21
Laser melting deposition [15]	400–480	14–16
SLM [10,19]	600–750	1–3

## 2. Materials and Experimental Procedures

### 2.1. Materials

The powder materials utilized in the SLM process were spherical tantalum powders with a purity of 99.9%, and the particle size ranged from 5 to 30 μm. The chemical composition of tantalum powders is presented in Table 2, and scanning electron microscopy (SEM) morphologies are shown in Figure 1a. The powders demonstrate a high degree of sphericalness and good flowability, with the particle size distribution illustrated in Figure 1b. The powders were subjected to a constant temperature of 80 °C for 6 h in a vacuum drying oven to eliminate any absorbed gases and moisture prior to the SLM process. The Ti-6Al-4V alloy served as the substrate for the fabrication process.

### 2.2. SLM Process

The SLM process was carried out using an EOS M290 device. The scanning strategy adopted in this study is presented in Figure 2a, utilizing a 67° angle for reversing the direction of scanning between the adjacent layers. The 67° rotation was adopted to achieve the maximum number of layers until the exact same scan vector orientation occurred [21]. This scanning strategy also weakens residual stresses and causes the smallest directional stress difference [22]. Argon was introduced into the device chamber to reduce the oxygen content to below 100 ppm before additive manufacturing. The macroscopic appearance of the prepared samples are shown in Figure 2b, which consists of 10 × 10 × 10 mm^3^ blocks and 40 × 6 × 9 mm^3^ rectangular blocks. The dimensions of the tensile test samples made from rectangular blocks is shown in Figure 2c. In this study, the layer thickness for the SLM process was 0.03 mm, and the scanning hatch distance was 0.07 mm, with the main process parameters presented in Table 3.

The study aimed to analyze the impact of volume energy density (*E*) on the microstructure and mechanical properties of tantalum, as expressed in equation [23]:*E = P/(v × h × t)*(1)
where *P*, *v*, *h*, and *t* represent the laser power [W], scan speed [mm/s], scan hatch distance [mm], and layer thickness [mm], respectively.

A vacuum heating treatment process was adopted to investigate the effect of heat treatments on the microstructure and mechanical properties of SLMed pure tantalum. The vacuum was pumped to below 10^−2^ Pa, and the SLMed tantalum samples were subjected to vacuum heat treatments at different temperatures of 1200 °C, 1400 °C, and 1600 °C with a heating rate of 10 °C/min and a holding time of 2 h, followed by cooling in a furnace.

### 2.3. Microstructural Characterization and Mechanical Properties

The samples were cut from the substrate using an electric spark wire cutting machine and then subjected to standard metallographic preparation procedures, including grinding and polishing, followed by 8 h of vibration polishing. The densities of the SLMed tantalum samples were measured by Archimedes’ method, with at least three measurements taken for each sample, and the average value was recorded. The relative density was used as a characterization of density, with the theoretical density of pure Ta being 16.68 g/cm^3^.

The microstructure and defects in the SLMed Ta samples were analyzed using SEM. The grain morphologies and textures were characterized using electron backscattered diffraction (EBSD) with a step size of 1 μm. The elemental impurity contents of oxygen and nitrogen were determined through LECO analyzers (OHN863).

The microhardness was measured using a Vickers microhardness tester with a diamond indenter with a load of 0.2 kgf and a dwell time of 15 s. Microhardness measurements were taken along the building direction at 0.5 mm intervals, as shown in Figure 2d, and the average value was recorded. Tensile tests were performed using a tensile testing machine, with the tensile direction appearing perpendicular to the building direction. Three samples were tested for each condition, and the average value was recorded. Finally, the fracture morphologies were analyzed using SEM after the tensile tests. The investigation process is shown in Figure 3.

## 3. Results

### 3.1. Density

Figure 4 illustrates the results of the relative density tests conducted on pure tantalum specimens fabricated with SLM at various energy densities. The microstructure morphologies of the specimens along the manufacturing direction were observed using SEM. The results show that the relative density of SLMed pure tantalum specimens initially increased significantly and then stabilized as the energy density rose. For instance, many pore defects were found in the internal pure tantalum prepared by SLM at an energy density of 342 J/mm^3^, with a size of about 100–120 μm, and some few spherical tantalum particles, which caused the relative density to decrease. This phenomenon was also observed in the SLM preparation of 304 L under a high energy density [24]. When the energy density was reduced to 190 J/mm^3^, the size of the pore defects decreased from 100 μm to about 30 μm. This reduction in defect size was attributed to the fact that the pore defects were induced by the splashing of the molten metal, and the degree of splashing was directly related to the amplitude of the laser energy density. Laser-splashing-induced pores are common to major defects in SLM-prepared pure tantalum. The relative density of pure tantalum prepared by SLM reached 99.9% at an energy density of 190 J/mm^3^, and no obvious unmelted spherical particles or microcracks were detected, indicating that the pure tantalum powders had undergone complete melting during the SLM process and the SLMed tantalum had good plastic formability.

### 3.2. Microstructure

Electron backscatter diffraction (EBSD) characterization was performed on the longitudinal section of tantalum to investigate the microstructure evolution of SLMed tantalum under different energy densities, as presented in Figure 5. Figure 5a,b displays the grain morphology and orientation, where high-angle grain boundaries (HAGBs) with misorientation angles greater than 10° are marked by black lines. It can be observed that the grains grew preferentially along the build direction as columnar grains [22], with irregular shapes and no identifiable single melt pool. The double-sided scanning strategy was adopted in the same layer, and the scanning direction of adjacent layers was rotated by 67°, resulting in the irregular shape of the grains. The orientation rotation phenomenon within some grains also indicated that the grain growth direction changed over time.

Low energy density results in an enhanced <110> texture along the transverse direction and increased dislocation density, as shown in Figure 5c,d. The main textures of SLMed tantalum made at a high energy density along the transverse direction are <100> and <110>. The trend of the <100> texture was reduced as the energy density decreased, while the trend of the <110> texture increased, which is similar to the texture distribution of the SLMed tantalum prepared by a 60° and 90° rotation scanning reported by Lore Thijs [13]. Energy density mainly affects the texture distribution by changing the shape of the melt pool and the direction and magnitude of local temperature gradients [25,26]. Figure 5c,d demonstrates that the grain size of SLMed tantalum prepared at different energy densities did not change significantly, but the dislocation density and the density of low-angle grain boundaries increased as the energy density decreased. Local misorientation changes can be used to qualitatively evaluate residual strain and dislocation density [27]. Figure 5e,f indicates that a low laser energy density could result in high local misorientations. This is because the higher the energy density, the more heat accumulation there is, which provides potential in situ annealing treatments, to reduce the dislocation density and strain inside the material [28,29]. Therefore, the high dislocation density formed under a lower laser energy density, could lead to the result of greater dislocation strengthening.

### 3.3. Chemical Composition Content

The oxygen and nitrogen contents of tantalum powders and SLMed tantalum with varying energy densities were measured to investigate the impact of different energy densities in selective laser melting on the dissolved oxygen and nitrogen contents, as presented in Table 4. It is evident from Table 4 that the oxygen and nitrogen contents increased with increasing energy density. The oxygen content of SLMed tantalum made at the energy density of 190 J/mm^3^ was elevated by 117 ppm compared to the tantalum powders, which resulted in an increase by about 37%. Meanwhile, the nitrogen content increased by 100 ppm, which indicates an increase by about 286%. The results indicate that the oxygen impurities mainly originated from tantalum powders, while nitrogen impurities mainly stemmed from the chemical reaction between molten tantalum and nitrogen in the atmosphere. The increase in oxygen and nitrogen contents with increasing energy density being elevated is attributed to the following two reasons: (1) the laser stays on the molten pool for a longer time at high energy densities, resulting in a slower non-equilibrium solidification rate and more time for oxygen and nitrogen diffusion into the tantalum samples; (2) the Marangoni convection of the molten metal is more pronounced, leading to a higher contact area between the splashed particles and the oxygen-nitrogen impurities in the forming chamber atmosphere.

### 3.4. Mechanical Properties

Tensile tests were conducted to investigate the impact of different energy densities on the mechanical properties of SLMed tantalum samples, and the results are presented in Figure 6. As depicted in Figure 6a, the ultimate tensile strength (UTS) of SLMed tantalum increased from 589 MPa to 640 MPa, and the fractured elongation from 8% to 16% as the energy density decreased from 342 J/mm^3^ to 190 J/mm^3^. The changes in the tensile properties of SLMed Ta under different energy densities are shown in Figure 6b,c. The results demonstrate that the tensile strength underwent three stages increasing, plateauing, and decreasing as the energy density decreased. Furthermore, the highest and lowest values of tensile strength were 640 MPa and 568 MPa, respectively. Meanwhile, the plasticity of the material continued to increase, with the highest and lowest values of fractured elongation being 16% and 8%, respectively. Notably, the optimal tensile performance of SLMed tantalum was achieved at 190 J/mm^3^, with a tensile strength of 640 MPa and a fractured elongation of about 16%.

### 3.5. Vacuum Heat Treatment Process

Tensile tests and microhardness tests were conducted to investigate the effect of the vacuum heat treatment at different temperatures on the mechanical properties of the SLMed tantalum samples are depicted in Figure 7. As shown in Figure 7a, the tensile strength of the SLMed tantalum decreased to 549 MPa, and the elongation increased to 28% after vacuum heat treatment at 1200 °C, indicating a significant improvement in toughness. The tensile strength decreased further as the vacuum heat treatment temperature increased while the plasticity increased. The tensile strength decreased to 345 MPa after vacuum heat treatment at 1600 °C, representing a 40.9% decrease compared to that before vacuum heat treatment, while the elongation was the highest, reaching 32%. The trend of microhardness change is consistent with that of tensile strength. The microhardness decreased almost linearly with the increase in vacuum heat treatment temperature, as shown in Figure 7b. The microhardness decreased from 248 HV0.2 to 135 HV0.2 after the vacuum heat treatment at 1600 °C, representing a decrease of about 45.6%.

## 4. Discussion

### 4.1. The Effect of Microstructure on Mechanical Properties

The ductility of SLMed tantalum was low at high energy densities due to the presence of large-sized pore defects that could cause local stress concentration, leading to crack initiation and the acceleration of the initial fracture during the tensile process. However, with the energy density being low, the local stress concentration effect was reduced significantly because of the small size and the low number of internal pore defects. Although small pore defects cannot be entirely eliminated by process optimization, they are not the main reason for the relatively low ductility of SLMed tantalum compared to powder metallurgy tantalum and electron beam melted tantalum.

Tensile tests were conducted, and the fracture morphologies were analyzed using SEM to investigate the influence of energy density on the ductility of SLMed tantalum. The fracture of SLMed tantalum manufactured at a high energy density (*E* = 342 J/mm^3^) was mainly intergranular and transgranular, as shown in Figure 8. The presence of unclosed large-sized pore defects severely weakened the ductility due to local stress concentrations, which led to crack initiation. This is consistent with the lower fracture elongation. In addition, Mode I opening failures [30] can be observed in Figure 8a. The fracture mechanism of SLMed tantalum is closely related to the segregation of oxygen atoms at grain boundaries, which are discussed in detail later. However, the fracture of SLMed tantalum prepared at a low energy density (*E* = 190 J/mm^3^) exhibited partial transgranular fracture and a small amount of dimple fracture with no Mode I opening failure, indicating a significant improvement in ductility. The reduced ductility of SLMed tantalum manufactured at a high energy density was attributed to large-sized pore defects, which could be suppressed by process optimization.

The texture proportion, grain boundary density (GBD), and average local misorientation (ALM) were statistically analyzed to deeply analyze the reason for the changes in the tensile strength of SLMed tantalum at different energy densities, as shown in Figure 9. The texture proportion analysis was performed to highlight the <100>, <110>, and <111> textures that were aligned with the transverse direction within 20° [9,13]. The grain boundary density (GBD) was represented by the total length of specified angle grain boundaries per unit area. The average local misorientation (ALM) was obtained by a weighted calculation of the local grain misorientation.

The statistical analysis revealed that the proportion of the <110> texture was higher along the transverse direction when *E* was 190 J/mm^3^, while the proportion of the <100> texture was relatively low. The <110> texture had higher strength due to the higher Taylor factor of the <110> texture than the <100> texture [13]. The density of large-angle grain boundaries (>10°) changed little with the change in energy densities, indicating that the irregular columnar grain size did not change significantly. However, the density of small-angle grain boundaries (2–10°) and dislocations density (<2°) first increased and then decreased with energy density being reduced. The average local misorientation also followed a pattern of first increasing and then decreasing. When *E* = 157 J/mm^3^, the density of small-angle grain boundaries and dislocations decreased, resulting in a reduction in dislocation slip resistance and a decrease in strength.

In conclusion, the microstructures mainly affecting the strength and toughness of pure tantalum prepared by SLM included pore defects, texture orientation, and dislocation density. The size and quantity of pore defects were significantly reduced but not completely eliminated as the energy density decreased from 340 J/mm^3^ to 190 J/mm^3^. At the same time, part of the <100> textures transformed into <110> textures, resulting in dislocation strengthening being enhanced and an increase in the tensile fracture strength.

### 4.2. The Effect of Oxygen and Nitrogen Content on Mechanical Properties

Both the powders and the forming chamber atmosphere contain oxygen and nitrogen impurities during the SLM additive manufacturing process. The production of pure tantalum using SLM requires attention to the source and the content of the oxygen-nitrogen impurities because tantalum’s mechanical properties are sensitive to impurities. The interstitial solid solution of oxygen and nitrogen could cause lattice distortion in the SLMed tantalum, resulting in dislocation pinning effects [31]. On one hand, this has a solid solution-strengthening effect. On the other hand, it significantly reduces toughness [15,32].

Table 5 presents the test results of oxygen and nitrogen variations in different additive manufacturing metals. It is evident that in both SLM and LMD processes, the content of oxygen and nitrogen impurities significantly increased compared with the metal powder used. In this study, the tantalum powders used had a low content of oxygen impurities. However, the oxygen content increased by about 37% after the SLM process at *E* = 190 J/mm^3^. This suggests that the oxygen impurities in the SLMed pure tantalum produced by SLM mainly came from the powder materials. That is, about 27% of the oxygen impurities came from the forming chamber atmosphere. Additionally, as shown in Table 5, the oxygen impurity content in pure tantalum produced by SLM was significantly higher than that of nitrogen impurities. In this study, when *E* was 190 J/mm^3^, the ratio of oxygen and nitrogen impurities in the tantalum samples was about 3:1. Nitrogen elements mainly came from the chemical reaction between molten tantalum in the SLM process and nitrogen in the forming chamber atmosphere. The relatively low content of oxygen and nitrogen impurities in pure tantalum produced at 190 J/mm^3^ in this study is one of the reasons for its high fracture elongation.

The microhardness results of pure tantalum was fabricated using SLM with various energy densities and are presented in Figure 10. As depicted in Figure 10a, the microhardness decreased with energy density being reduced, primarily due to a significant reduction in the oxygen impurity content (Table 4). As illustrated in Table 4 and Figure 6c, when the energy density was reduced from 310 J/mm^3^ to 256 J/mm^3^, the oxygen and nitrogen contents decreased by 1.4% and 23%, respectively, and the fractured elongation increased by 17%. When the energy density was further reduced from 256 J/mm^3^ to 157 J/mm^3^, the oxygen and nitrogen contents decreased by about 15% and 4%, respectively, and fractured elongation further increased by 3%. Therefore, the fractured elongation of tantalum fabricated by SLM was more sensitive to nitrogen impurities compared with oxygen.

The interstitial solid solution of oxygen and nitrogen could significantly affect the yield strength of pure tantalum [15]. The yield strength (*YS*) of pure tantalum can be expressed as follows [32]:*YS* = 123.48 + 0.875 × *C_N_* + 0.444 × *C_O_*(2)
where *C_N_* and *C_O_* are the contents of dissolved nitrogen and oxygen, respectively. The unit of *C_N_* and *C_O_* is ppm. According to the linear relationship between the microhardness and yield strength [36], the microhardness (MH) of tantalum could be expressed as follows:*MH* = 0.57 × (123.48 + 0.875 × *C_N_* + 0.444 × *C_O_*)(3)

When the energy density is low, the calculated value of *MH* is close to the measured value, as demonstrated in Figure 10a. However, as the energy density increases, the calculated value of *MH* gradually becomes significantly higher than the measured value. This is because oxygen impurities tend to segregate at grain boundaries at high energy densities (Figure 10b), although the oxygen impurity content increases, which would weaken the solid solution strengthening effect. Moreover, the segregated oxygen impurities could reduce the bonding strength of grain boundaries, leading to intergranular and Mode I fracture failures.

### 4.3. The Effect of Vacuum Heat Treatments on Mechanical Properties

The density of SLMed tantalum remained unchanged after vacuum annealing at 1600 °C, but the fractured elongation increased from 18% to 32% (Figure 7a), which is comparable to that of powder metallurgy fabricated tantalum. The microstructure of SLMed pure tantalum was analyzed using EBSD to analyze the effect of vacuum annealing on toughness, as shown in Figure 11. It can be observed that the proportion of the <110> texture decreased after vacuum annealing, while the proportion of the <100> texture increased. This indicates that recrystallization occurred in SLMed pure tantalum with the vacuum heat temperature exceeding the recrystallization temperature (about 1100 °C) [18,37]. The grain size of SLMed pure tantalum significantly increased after vacuum annealing at 1600 °C, as shown in Figure 11b. However, the <100> texture proportion increased significantly after vacuum annealing at 1200 °C, but the tensile strength only decreased by about 14% (Figure 7a). Therefore, texture orientation was not the main factor affecting the strength of SLMed pure tantalum.

Figure 12 shows the calculated results of the effects of different vacuum annealing temperatures on the GBD and ALM of SLMed tantalum. It can be seen that GBD and ALM decreased by 76% and 67% after vacuum annealing at 1200 °C, respectively. GBD and ALM decreased significantly further as the vacuum annealing temperature increased to 1400 °C or 1600 °C. The decrease in GBD and ALM indicated a decrease in deformation dislocation slip resistance, which, on one hand, led to a continuous decrease in the tensile strength and microhardness, and on the other hand, increased the fractured elongation.

The deformation-induced dislocation could be hindered by oxygen and nitrogen impurities, small-angle grain boundaries, and ALM during the tensile test. Dislocations tend to pile up at high-density small angle grain boundaries inside grains, leading to local stress concentrations. When the deformation stress exceeds the binding strength, a transgranular fracture could occur (Figure 8c,d). Transgranular cleavage fractures transformed into ductile dimple fractures with an obvious necking phenomenon after the vacuum heat treatment at 1600 °C (Figure 13a,b). This is because the decrease in GBD and ALM could lead to a decrease in dislocation slip resistance and an increase in the dislocation mean free path. Dislocations tend to pile up in oxygen impurities. When the deformation stress exceeds the binding strength of oxygen impurities, voids aggregate, and then ductile dimple fracture occurs.

## 5. Conclusions

This study analyzed the microstructure and strengthening-toughing mechanism of SLM tantalum. The main conclusions are as follows:(1)Pure tantalum with a relative density as high as 99.9% was fabricated by selective laser melting additive manufacturing technology. The content of pore defects and oxygen-nitrogen impurities significantly decreased as the energy density decreased from 342 J/mm^3^ to 190 J/mm^3^, leading to an increment in the fractured elongation by about 100%.(2)The ratio of oxygen and nitrogen impurities in SLMed tantalum was about 3:1, meaning that oxygen impurities were the major impurities. The oxygen impurities mainly result from tantalum powders, while most nitrogen impurities are from the chemical reaction between the molten liquid tantalum and nitrogen in the forming chamber atmosphere. The fractured elongation of SLMed tantalum is more sensitive to nitrogen impurities than oxygen. The oxygen-nitrogen solid solution significantly enhances the microhardness of SLMed tantalum, with the energy density being 190 J/mm^3^. However, the oxygen-nitrogen solid solution strengthening is weak due to the oxygen segregation at the grain boundaries, with the energy density being 310 J/mm^3^.(3)The proportion of the <110> texture decreases after vacuum heat treatments with the temperature being higher than 1200 °C. Moreover, the densities of dislocations and small-angle grain boundaries are significantly reduced. The fractured elongation of SLMed tantalum prepared at 190 J/mm^3^ increases to 28%, and the strength decreases by about 14% after vacuum heat treatment at 1200 °C.

Reducing the content of oxygen impurities and heat treatment can effectively improve the plasticity of SLMed tantalum. While using tantalum powder with a low oxygen and nitrogen impurity content, it is possible to prepare high toughness and strength SLMed tantalum by adding small amounts of reducing components to the powders in the future. In addition, a novel scanning strategy, such as laser polishing, may improve the tensile mechanical properties by decreasing the dislocation density.

## Figures and Tables

**Figure 1 materials-16-03161-f001:**
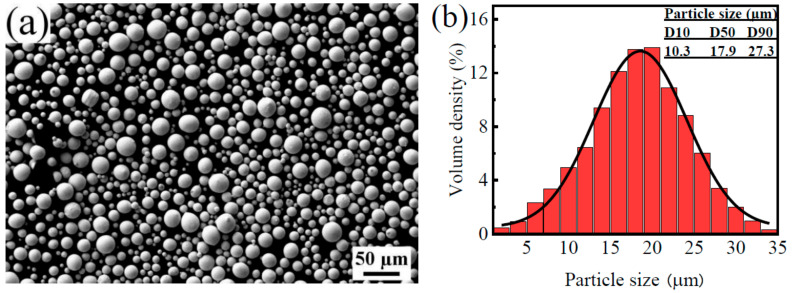
(**a**) Morphologies and (**b**) Particle size distributions of Ta powders.

**Figure 2 materials-16-03161-f002:**
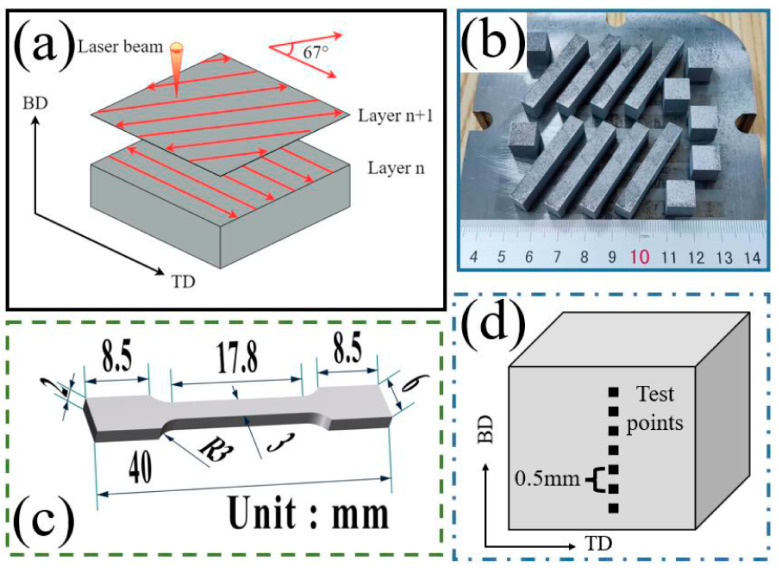
(**a**) The scan strategy schematic diagram of the manufacturing process, (**b**) The tantalum parts manufactured by SLM, (**c**) The schematic diagram of the sample size for tensile test, and (**d**) the schematic diagram of the hardness test positions. (Note: BD is the building direction, while TD is the transverse direction).

**Figure 3 materials-16-03161-f003:**
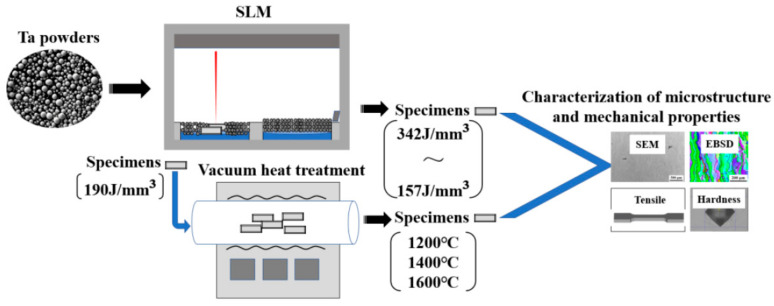
Schematic illustration of the investigation process.

**Figure 4 materials-16-03161-f004:**
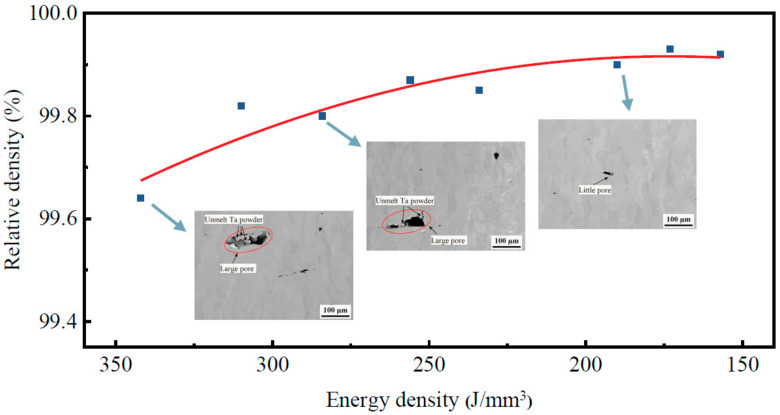
The relative density results of SLMed Ta made with different energy densities and corresponding SEM images along the building direction.

**Figure 5 materials-16-03161-f005:**
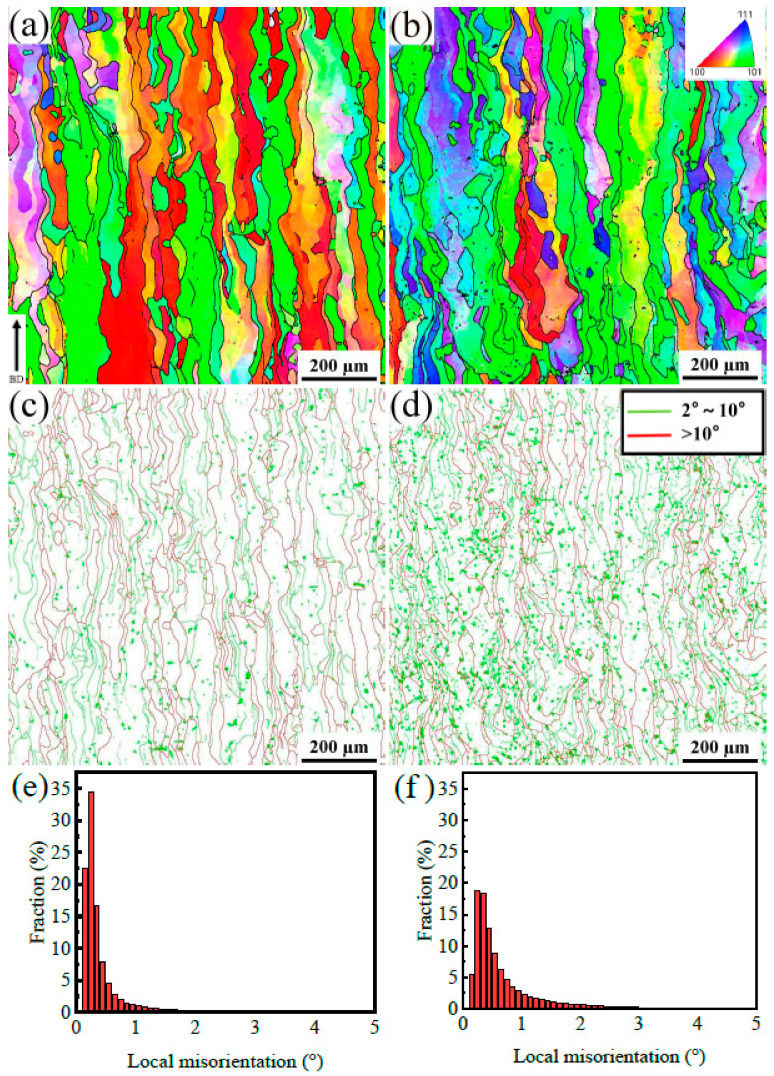
The EBSD results along the building direction showing the grain morphology, grain boundary and local misorientation distribution of SLMed Ta manufactured at different energy densities of (**a**,**c**,**e**) 342 J/mm^3^ and (**b**,**d**,**f**) 190 J/mm^3^, respectively.

**Figure 6 materials-16-03161-f006:**
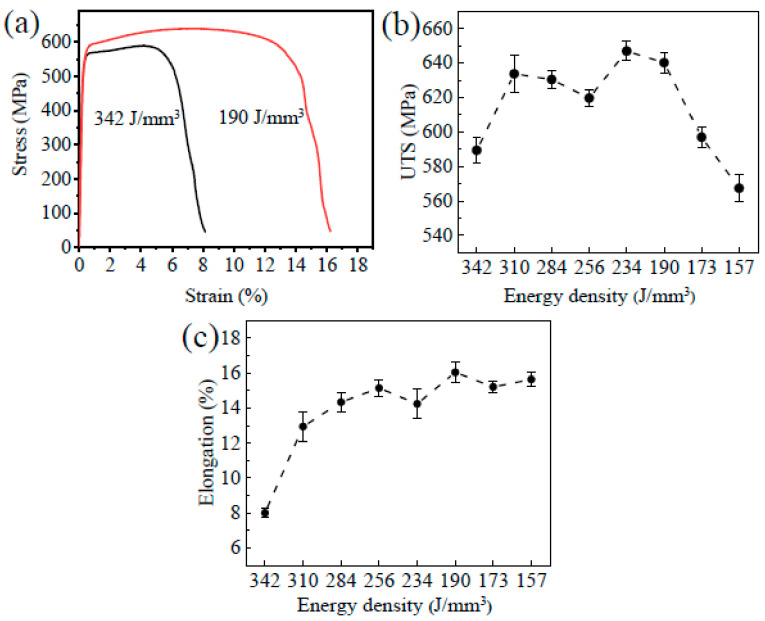
(**a**) Tensile stress–strain curves, (**b**) Ultimate tensile strength and (**c**) Elongation of SLMed Ta prepared at different energy densities.

**Figure 7 materials-16-03161-f007:**
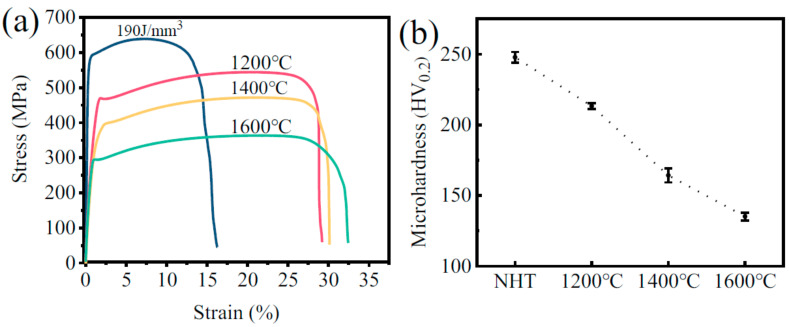
(**a**) Tensile stress–strain curves and (**b**) Microhardness of SLMed Ta made at 190 J/mm^3^ after heat treatments of different temperature. (Note: NHT is the SLMed Ta prepared at 190 J/mm^3^ without heat treatments.).

**Figure 8 materials-16-03161-f008:**
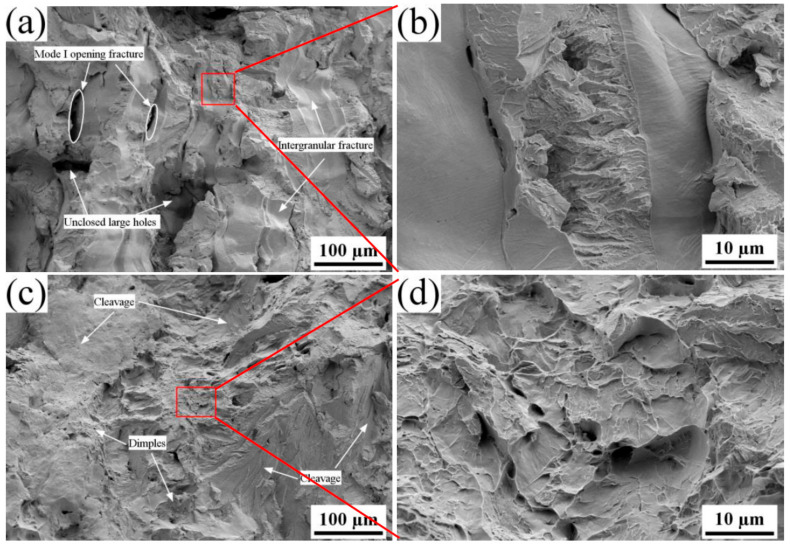
Fracture morphologies of SLMed Ta manufactured at different energy densities: (**a**,**b**) 342 J/mm^3^ and (**c**,**d**) 190 J/mm^3^.

**Figure 9 materials-16-03161-f009:**
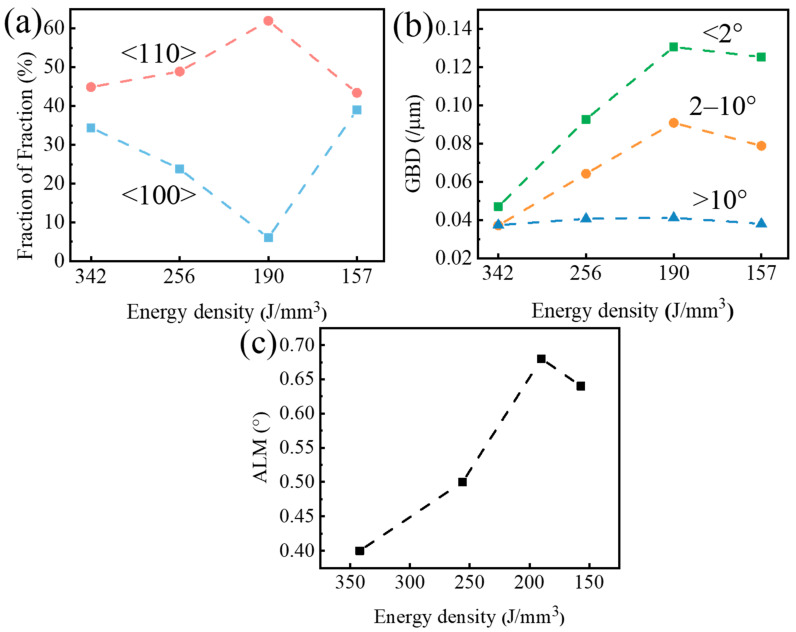
(**a**) Fraction of texture components, (**b**) Grain boundary densities, and (**c**) Average local misorientation of SLMed Ta made by different energy densities.

**Figure 10 materials-16-03161-f010:**
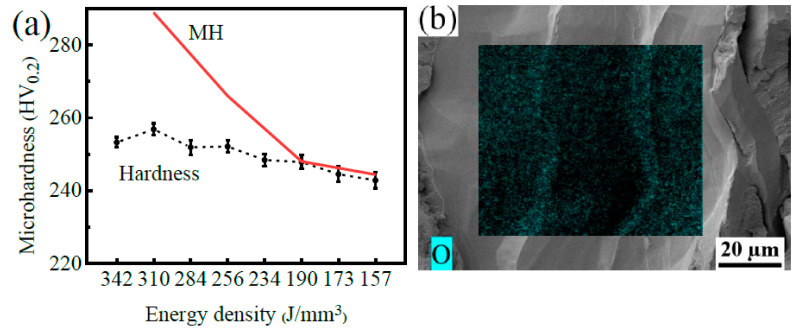
(**a**) The microhardness and *MH* results of SLMed Ta fabricated at different energy densities and (**b**) the oxygen impurity (green dots shown in the figure) distribution on the intergranular fracture of SLMed Ta made at 310 J/mm^3^.

**Figure 11 materials-16-03161-f011:**
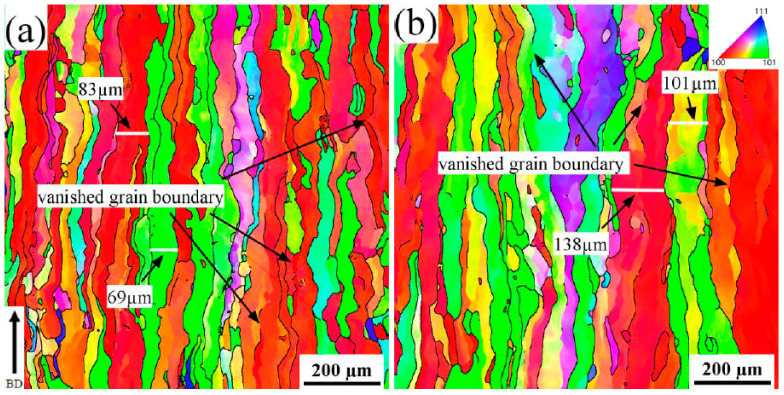
The EBSD results along the building direction of SLMed Ta after heat treatments with different temperature: (**a**) 1200 °C and (**b**) 1600 °C.

**Figure 12 materials-16-03161-f012:**
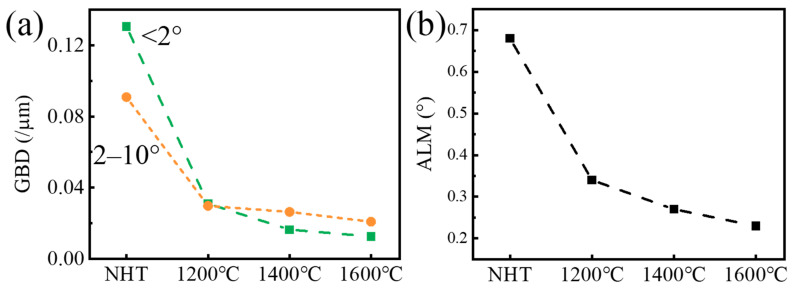
(**a**) Grain boundary densities and (**b**) Average local misorientation after post heat treatments.

**Figure 13 materials-16-03161-f013:**
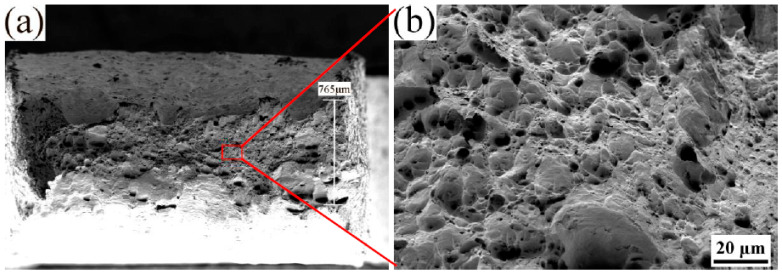
(**a**) Fracture morphologies of SLMed Ta made at 190 J/mm^3^ after 1600 °C heat treatments and (**b**) The enlarged image of the tough fracture within the red box.

**Table 2 materials-16-03161-t002:** Chemical composition of pure tantalum powder materials.

Element	Ta	O	C	Mn	K	Total Impurities
wt%	Balance	0.0317	0.0088	0.05	0.0013	<0.1

**Table 3 materials-16-03161-t003:** Process parameters and energy densities of SLMed Ta.

Sample	Scanning Speed (mm/s)	Laser Power (W)	Energy Density (J/mm^3^)
1	390	280	342
2	430	280	310
3	470	280	284
4	520	280	256
5	570	280	234
6	700	280	190
7	770	280	173
8	850	280	157

**Table 4 materials-16-03161-t004:** Contents of oxygen and nitrogen impurities in the raw powders and the SLMed Ta. (Unit: ppm).

Sample	Oxygen	Nitrogen
Ta powders	317	35
157 J/mm^3^	426	132
190 J/mm^3^	434	135
256 J/mm^3^	501	137
310 J/mm^3^	508	179

**Table 5 materials-16-03161-t005:** The content of impurities and mechanical properties with different manufacturing techniques and metal powders. (Elemental content in ppm).

Process	Material	Powder		As-Built Part		UTS (MPa)	Elongation (%)
		Oxygen	Nitrogen	Oxygen	Nitrogen		
SLM *	Ta	317	35	434	135	640	16
LMD [15]	Ta	405	180	700	500	430	14
SLM [33]	Ta	829	11	1422	23	660	4
SLM [34]	Steel	541	181	569	368	1440	1.5
SLM [35]	Ti-6Al-4V	1308	62.2	1690	124	1213	8.6

* This research.

## Data Availability

The data and results involved in this study have been presented in detail in the paper.

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
