# Peer review of "Selective Laser Melting Additive Manufactured Tantalum: Effect of Microstructure and Impurities on the Strengthening-Toughing Mechanism"

_materials, 2023, doi:10.3390/ma16083161_

Round 1

Reviewer 1 Report

There are some questions need authors to answer:

1. It seems that Fig. 3 and Fig. 4 show the same results (high energy density, high porosity). I would like to ask why the authors show density and porosity at the same time in this paper.

2. In Fig. 5, if the density of subgrain(<10 degrees) in Fig. 5(d) is dense than Fig. 5(c), why the Fig. 5(f) shows less grains

3. In the paragraph of line208, "This is because the higher the energy density, the more heat accumulation there is, which provides potential in-situ annealing treatment, to reduce the dislocation density and strain inside the material." Do you have any references to support this statement? The subgrain density should relate to the cooling rate of the melt pool.

4. How many samples do you test in the tensile test? You should include the standard deviation of UTS and Elongation in Fig. 6(b), like in Fig. 7(b)?

Author Response

We sincerely thank you for the constructive comments and suggestions that are so valuable in improving the quality of our manuscript. The comments have been presented below followed by our responses. The modified part in the revised manuscript has been marked in red font.

1) It seems that Fig. 3 and Fig. 4 show the same results (high energy density, high porosity). I would like to ask why the authors show density and porosity at the same time in this paper.

Answer

       Indeed, both figures show the same results. We wanted to prove that the distribution of surface defects and volume defects is consistent in the past and that there is no segregation problem of a large number of hole defects. But we decided to delete Figure 4 after careful consideration based on the result.

2) In Fig. 5, if the density of subgrain(<10 degrees) in Fig. 5(d) is dense than Fig. 5(c), why the Fig. 5(f) shows less grains?

Answer

       We apologize for the lack of clarity. Figure 4(e-f) (after initial Figure 4 was deleted) is not the grain boundary distribution. Figure 4(e-f) shows the local misorientation (0-5°) inside the grain, and the unit is the percentage. The local misorientation inside the grains shifts to a large angle with the high density of subgrains.

3) In the paragraph of line 208, "This is because the higher the energy density, the more heat accumulation there is, which provides potential in-situ annealing treatment, to reduce the dislocation density and strain inside the material." Do you have any references to support this statement? The subgrain density should relate to the cooling rate of the melt pool.

Answer

The references [28,29] pronounced that the increased energy density can result in the thermalization of the energy and the thermal accumulation is equivalent to preheating the previous fabricated part which can decrease the residual stress. The paper [R1] explained that grain coarsening is caused by thermal accumulation and overheating at high energy density. The research [R2] shows that due to the significant thermal accumulation and high temperature within the molten pool at high energy density, the effect of quenching that occurs by conduction of heat through the substrate is not significant in this situation, favoring the complete transformation of β → α phase after solidification. These support this statement.

4) How many samples do you test in the tensile test? You should include the standard deviation of UTS and Elongation in Fig. 6(b), like in Fig. 7(b)?

Answer

       We appreciate the reviewer’s suggestion. Three samples were tested for each condition. We have updated Figure 5 (after Figure 4 was deleted) with the standard deviation of UTS and Elongation.

Reviewer 2 Report

The authors studied the selective laser melting additive manufactured tantalum: effect of microstructure and impurities on the strengthening-toughing mechanism. The manuscript had an interesting topic and was well-written, however it could only be accepted with the following minor revisions:

1.    SLM was a type of additive manufacturing (AM) technology. It is advised at the beginning of the Introduction Section the authors should define briefly AM technology, types and its advantages. Therefore, it is recommended the authors can add the following papers as references; Investigation of ABS–oil palm fiber (Elaeis guineensis) composites filament as feedstock for fused deposition modeling, Rapid Prototyping Journal, 2022, (https://doi.org/10.1108/RPJ-05-2022-0164); Effect of HBN fillers on rheology property and surface microstructure of ABS extrudate, Jurnal Teknologi, 84(4), 175-182.

2.    Please check, the text didn't define Ta's acronym.

3.    “The densities of the SLMed tantalum samples were measured by Archimedes’ method”, which scientific studies did you cite?

4.    For the experimental procedure, it is recommended the authors provide a flow chart for the experimental setup.

5.    Section 3.1, 3.3, and 3.4 was very poor in citing the previous studies to compare with the result of the present study.

6.    Figure 6(b), the graph of UTS and elongation must be shown together with error bars indicating the standard deviation for each sample.

7.    The conclusion could be improved by adding future studies, limitations, and implications for researchers.

8.    38% of the references were outdated (past 5 years). The most recent reference must be updated accordingly throughout the paper.

Author Response

We sincerely thank you for the constructive comments and suggestions that are so valuable in improving the quality of our manuscript. The comments have been presented below followed by our responses. The modified part in the revised manuscript has been marked in red font.

1) SLM was a type of additive manufacturing (AM) technology. It is advised at the beginning of the Introduction Section the authors should define briefly AM technology, types and its advantages. Therefore, it is recommended the authors can add the following papers as references; Investigation of ABS–oil palm fiber (Elaeis guineensis) composites filament as feedstock for fused deposition modeling, Rapid Prototyping Journal, 2022, (https://doi.org/10.1108/RPJ-05-2022-0164); Effect of HBN fillers on rheology property and surface microstructure of ABS extrudate, Jurnal Teknologi, 84(4), 175-182.

Answer

       Many thanks for the kind suggestion. The brief AM technology and the advantages of SLM are added in the paragraph of lines 37-42.

2) Please check, the text didn't define Ta's acronym.

Answer

We feel very sorry for our carelessness. We have defined Ta's acronym in the paragraph of lines 13 and 30.

3) “The densities of the SLMed tantalum samples were measured by Archimedes’ method”, which scientific studies did you cite?

Answer

Additive manufacturing metals generally use this method to measure density. It has been reported in the ref. [7], [11] and [14] in which this method is used to measure the SLMed tantalum.

4) For the experimental procedure, it is recommended the authors provide a flow chart for the experimental setup.

Answer

Thank for the reviewer’s suggestion. The schematic diagram of experimental setup has been provided in the response letter.

Figure R1 The schematic diagram of study process

5) Section 3.1, 3.3, and 3.4 was very poor in citing the previous studies to compare with the result of the present study.

Answer

We are sorry for not fully comparing the present studies. Because few studies on selective laser melting of tantalum in recent years, there is not much data that can be cited and the cited data about SLMed tantalum has already been written in the introduction part.

6) Figure 6(b), the graph of UTS and elongation must be shown together with error bars indicating the standard deviation for each sample.

Answer

We appreciate the reviewer’s suggestion. We have updated Figure 5 with the standard deviation of UTS and Elongation.

7) The conclusion could be improved by adding future studies, limitations, and implications for researchers.

Answer

We are grateful for the reviewer’s suggestion. We added the future study ideas in the paragraph of lines 447-452.

8) 38% of the references were outdated (past 5 years). The most recent reference must be updated accordingly throughout the paper.

Answer

We appreciate the reviewer’s carefulness and patient. We have tried our best to update fourteen references.

Reviewer 3 Report

In this paper, pure Ta specimens manufactured by SLM process was investigated experimentally. The effect of energy density on the relative densities of SLM tantalum samples was investigated. The paper has a high scientific level, materialized in important applied contributions that have a character of originality.

The content of the paper is good, but the suggestions are the following:

1. In the introduction, references should be added regarding heat treatments (classic and concentrated solar energy) for specimens manufactured by SLM process from Ti-6Al-4V alloy.

2. What is the reason why the authors positioned on angle of 67° specimens?

3. The variation in hardness should be explained in more detail. Why does it vary? What are the causes? Microhardness Tests - the explanations on microhardness testing should be supplemented by the indication of the test positions (e.g. graphically).

4. The statistical analysis of the experimental results (tensile, microhardness) of this paper should been added and discussed (mean, standard deviation, CV). What standard was used for the tensile tests? How many specimens were tested?

Author Response

We sincerely thank you for the constructive comments and suggestions that are so valuable in improving the quality of our manuscript. The comments have been presented below followed by our responses. The modified part in the revised manuscript has been marked in red font.

1) In the introduction, references should be added regarding heat treatments (classic and concentrated solar energy) for specimens manufactured by SLM process from Ti-6Al-4V alloy.

Answer

       We feel thankful for the reviewer’s opinion. We have added the content about the heat treatments of SLMed Ti-6Al-4V alloy by concentrated solar energy in the paragraph of lines 80-90.

2) What is the reason why the authors positioned on angle of 67° specimens?

Answer

       We feel sorry about misusing. The 67° rotation was adopted to achieve the maximum number of layers until the exact same scan vector orientation occurs. This scanning strategy will weaken the residual stresses and cause the smallest directional stress difference. The 67° rotation is the most commonly used rotation in selective laser melting. We have added this explanation in the paragraph of lines 122-125. (Please refer to Ref. [21] & [22])

3) The variation in hardness should be explained in more detail. Why does it vary? What are the causes? Microhardness Tests - the explanations on microhardness testing should be supplemented by the indication of the test positions (e.g. graphically).

Answer

       Many thanks for the kind suggestion. The hardness is mainly influenced by the distribution and content of oxygen-nitrogen impurities before heat treatment, while it is mainly affected by the microstructures such as dislocation density, low angle grain boundary density, etc. These have been explained in the manuscript. We have altered Figure 2 with the schematic diagram of the hardness test positions.

4) The statistical analysis of the experimental results (tensile, microhardness) of this paper should been added and discussed (mean, standard deviation, CV). What standard was used for the tensile tests? How many specimens were tested?

Answer

We feel thankful for the reviewer’s opinion. We analyzed the mean value of the tensile and microhardness tests in the manuscript. We also wanted to discuss the standard, but we didn’t find an obvious law for the standard deviation. GB/T228.1-2010 standard was used for the tensile tests. Three samples were tested for each condition. We have updated the Figure 5 with the standard deviation of UTS and Elongation.

Reviewer 4 Report

The manuscript discusses the influence of input energy density on the microstructure and impurities, and corresponding mechanical properties. Further, a vacuum heating treatment process was used to investigate the effect of heat treatments on the microstructure and mechanical properties of the fabricated specimens. The amount of oxygen and nitrogen impurities rises with an increase in the input energy density. The results showed that the enhancement of the hardness at 190 J/mm3 is due to the oxygen-nitrogen solid solution. At higher values of the input energy density, the solid solution strengthening effect is weaker due to the segregation of the oxygen at the grain boundaries. The heat treatment of the specimen manufactured by an input energy density at temperatures higher than 1200 o C leads to a decrease in the <110> texture and dislocation density. The fracture elongation rises by 28% after the application of heat treatment at a temperature of 1200o C.

Overall, the paper is very interesting and could be very useful for the readers. I recommend this manuscript be published in the present form.  

Author Response

 Thank you very much for your approval. We are thrilled to hear that you find our paper interesting and potentially useful for readers. We appreciate your time and effort in reviewing our manuscript. We have carefully considered the comments and made revisions accordingly. We hope that the changes we have made address all your concerns about the article and improve its quality. Once again, thank you for your valuable input and support.

Round 2

Reviewer 2 Report

Figure R1 "The schematic diagram of study process" (in the response letter) should be included in the manuscript.

Author Response

We feel sorry about our carelessness. We have added the schematic diagram of investigation process in the manuscript.